# MSX1—A Potential Marker for Uterus-Preserving Therapy of Endometrial Carcinomas

**DOI:** 10.3390/ijms21124529

**Published:** 2020-06-25

**Authors:** Simon Eppich, Christina Kuhn, Elisa Schmoeckel, Doris Mayr, Sven Mahner, Udo Jeschke, Julia Gallwas, Helene Hildegard Heidegger

**Affiliations:** 1Department of Obstetrics and Gynecology, University Hospital, Ludwig Maximilians University (LMU), Marchioninistraße 15, 81377 Munich, Germany; simon.eppich@gmx.de (S.E.); christina.kuhn@med.uni-muenchen.de (C.K.); sven.mahner@med.uni-muenchen.de (S.M.); julia.gallwas@med.uni-muenchen.de (J.G.); Helene.heidegger@med.uni-muenchen.de (H.H.H.); 2Department of Pathology, LMU Munich, Thalkirchner Str. 56, 80337 Munich, Germany; elisa.schmoeckel@med.uni-muenchen.de (E.S.); doris.mayr@med.uni-muenchen.de (D.M.); 3Department of Obstetrics and Gynecology, University Hospital Augsburg, Stenglinstr. 2, 86156 Augsburg, Germany; 4Department of Gynecology and Obstetrics, Georg August University Goettingen, University Medicine, 37075 Goettingen, Germany

**Keywords:** MSX1, endometrial carcinoma, cell-cycle, DNA-methylation, uterus-preserving therapy

## Abstract

Prognostic factors are of great interest in patients with endometrial cancer. One potential factor could be the protein MSX1, a transcription repressor, that has an inhibitory effect on the cell cycle. For this study, endometrioid endometrial carcinomas (*n* = 53), clear cell endometrial carcinomas (*n* = 6), endometrioid ovarian carcinomas (*n* = 19), and clear cell ovarian carcinomas (*n* = 11) were immunochemically stained for the protein MSX1 and evaluated using the immunoreactive score (IRS). A significant stronger expression of MSX1 was found in endometrioid endometrial carcinomas (*p* < 0.001), in grading 2 (moderate differentiation) (*p* = 0.001), and in tumor material of patients with no involvement of lymph nodes (*p* = 0.031). Correlations were found between MSX1 expression and the expression of β-Catenin, p21, p53, and the steroid receptors ERα, ERβ, PRα, and PRβ. A significant (*p* = 0.023) better survival for patients with an MSX1 expression in more than 10% of the tumor cells was observed for endometrioid endometrial carcinomas (21.3 years median survival (MSX1-positive) versus 17.3 years (MSX1-negative)). Although there is evidence that MSX1 expression correlates with improved long-term survival, further studies are necessary to evaluate if MSX1 can be used as a prognostic marker.

## 1. Introduction

Worldwide, endometrial cancer (EC) has become the 7th most common malignancy within the female population due to its incidence of 142,000 new cases per year [1]. Especially women in North America and Western Europe are affected [1]. In Germany, EC is the fifth most common malignancy of the female population and even the most common one within the group of malignancies of the female genitalia [2]. Until the age of 84, morbidity is continuously rising throughout the population [2].

However, this does not mean that only elderly women are affected. A total of 15% of ECs are diagnosed in premenopausal women who are aged 45 or younger and 4% in women who are even younger than 40 [3,4]. This leaves us with a group of EC patients who might still want to give birth and, therefore, would benefit from a fertility-preserving therapy.

In early EC, hysterectomy still is first-line treatment, leaving most of the patients cured (almost 100%) but infertile [1]. According to a meta-analysis by Gallos et al. the regression rate of those patients, who received fertility-sparing therapy, lies at 76%, but the relapse rate during follow-up is about 41% [5]. Also, an upgrade of disease was reported in some cases [5].

According to Bokhman, endometrial carcinomas can be classified into two groups: Type 1 tumors are more common and usually estrogen-dependent low-grade endometrioid carcinomas, while the less common type 2 tumors are usually estrogen-independent and contain high-grade serous or clear-cell carcinomas [6,7].

For this study, we examined endometrioid and clear-cell carcinomas of the endometrium. We also included clear-cell and endometrioid carcinomas of the ovary, hoping to get more insight into these very rare tumor-subtypes. In a less accepted paradigm, ovarian tumors can also be classified as type 1 and type 2 tumors: endometrioid and clear-cell carcinomas are, among low-grade serous carcinoma, mucinous, and malignant Brenner tumors, classified as type 1 tumors, while type 2 tumors consist of high-grade serous carcinoma, carcinosarcoma, and undifferentiated carcinoma [7,8].

The combination of these types of carcinoma has the advantage that the corresponding subtypes can easily be compared because the three-staged tumor grading-system of the endometrioid ovarian carcinoma is equivalent to the grading-system of the endometrioid endometrial carcinoma, while clear-cell carcinomas of both, the ovary and the endometrium, are categorized as high-grade carcinomas [9]. According to several recent studies both subtypes of ovarian carcinomas can arise from atypical endometriosis that is being moved to the ovary by retrograde menstruation [10,11,12]. So, even though the tumorigenesis may be different, whether it is a tumor of the ovary or of the endometrium, the tumor biology can be compared and, by this, a larger group of patients gained.

For those younger patients, who want to preserve their fertility, it is important to have a precise knowledge about the course of disease, so that an oncologic security can be granted. Hence, this paper’s aim is to find a specific prognostic factor for a better survival in endometrial cancer patients and try to find out a marker for a better decision for or against a uterus-sparing therapy in patients with endometrial cancer with a high oncologic security.

The gene *MSX1* (Msh homeobox 1) is located on chromosome 4p16.2 and encodes the homonymous protein MSX1. The protein acts as a transcriptional repressor during embryogenesis and is thought to play part in the formation of limb-patterns, craniofacial development, odontogenesis, and tumor growth inhibition [13,14,15,16,17]. Also a number of diseases, like Witkop-Syndrome and non-syndromic tooth agenesis, has been linked to mutations in the *MSX1* gene [18,19]. Furthermore, the *MSX1* gene can be affected by the congenital Wolf-Hirschhorn-Syndrome leading to oligodontia [20]. In earlier studies, aberrant methylation of MSX1 promoter DNA has been linked to lung cancer, gastric cancer, Wilms tumor, childhood acute T-lymphoblastic leukemia, and breast cancer [21,22,23,24,25].

So we decided to investigate the protein MSX1, which is reported to have a regulating influence on the cell cycle and which plays an important role in the tumorigenesis of endometrial cancer [26,27].

## 2. Results

### 2.1. MSX-1 Staining in Endometrial and Ovarian Carcinomas

To evaluate staining, we used the IRS (immunoreactive score), which is the result of the percentage of stained carcinoma cells (0 = 0%, 1 = 1–10%, 2 = 11–50%, 3 = 51–80%, 4 ≤ 81%) multiplied with the coded staining intensity (0 = no staining, 1 = weak intensity, 2 = moderate intensity, 3 = strong intensity) [28].

Type of carcinoma showed a median IRS of 1 in endometrioid endometrial carcinomas (with a box length of 0–2 and a 95th percentile at 4), while endometrioid ovarian carcinomas and the clear cell type carcinomas had a median IRS of 0 (*p* < 0.001) (Figure 1).

Another significant difference was found in grading (grading 1: well-differentiated; grading 2: moderately differentiated; grading 3: badly differentiated): The highest IRS was found in graded 2 carcinomas (median IRS of 1 with a box length of 0–3.5 and a 95th percentile at 8), while graded 1 and 3 carcinomas showed a median IRS of 0 (*p* = 0.001) (Figure 2).

The tumor status (t-status) showed significant differences for the MSX1-staining in T1, T2, and T3 tumors (*p* = 0.03). T2 and T3 tumors showed a median IRS of 0. For T1 tumors the median IRS is 0 with a box length of 0–1 and a 95th percentile at 2 (Figure 3).

Another significant difference in the IRS score was found for the N status (lymph node staging). Those patients, who had tumor-infiltrated lymphoid nodes showed a median IRS of 0. On the other side, patients with no infiltrated lymphoid nodes had a median IRS of 0 with a box length of 0–1.5 and a 95th percentile at 2 (*p* = 0.031) (Figure 4).

No significant differences were found for FIGO-staging and metastasis status (M). For FIGO-staging the *p*-value was 0.31. The IRS was mostly 0 (78% for FIGO 1; 72% for FIGO 2; 62% for FIGO 3; 61% for FIGO 4). Information on metastasis status was provided in only two cases. Both cases were metastasis positive and had an IRS of 0 (These and other non-significant results can be found in the Appendix A).

### 2.2. MSX-1 Epression Regarding Survival

A significant better survival over the whole study cohort was found in those cases, with more than 10% of the tumor cells that are showing a positive staining for MSX1 (*p* = 0.023). This was the case especially in the subgroup of patients with endometrioid endometrial carcinomas. Of those patients, 100% were still alive 10 and even 15 years post-surgery. For those patients, the median survival was 21.3 years whereas the subgroup with less than 10% MSX1-expression or no expression at all had a median survival of 17.3 years (Figure 5). The reason for this cut-off was the percentage distribution of type of tumor. When looking at endometrioid carcinomas the boxplot displaying the 0–50th percentile is located at the 10% mark.

### 2.3. Multivariate Analysis

For multivariate analysis, we used Cox regression with tumor-related death as an endpoint. Our findings do not show any significant results. The lowest *p*-value can be found for the variable “MSX1 > 10%” (MSX1-expression in more than 10% of tumor cells) (*p* = 0.066) with a hazard ratio of 0.147. The exact results are listed in Table 1.

### 2.4. Correlation Analysis of MSX-1 with Other Parameters

In our research group, the same collective of patients has already been worked with. In the corresponding papers, the expression of ARID1A, β-Catenin p16, p21, p53, and the steroid receptors ERα, ERβ, PRα, and PRβ has been described [29,30]. In these studies, tumor material of the exact same patients has been immunohistochemically stained for the above-mentioned proteins and respective expression analyzed. These data can be compared with the results of our current study. This allows us to test for correlations between MSX1 and the above-mentioned proteins. Significant correlations were found between MSX1 and β-Catenin, p53, p21, ERα, ERβ, PRα, and PRβ while none were found between MSX1 and ARID1A or p16. The exact results are listed in Table 2.

## 3. Discussion

With the cell cycle having an important role in endometrial cancer and MSX1 inducing G0/G1 arrest, we figured MSX1 to be a positive prognostic factor for survival of patients with this sort of tumor. For this reason, we analyzed the expression of MSX1 in endometrioid and clear cell carcinomas of the endometrium and the ovary.

Jerzak et al. also showed in their paper the importance of the regulation of the cell-cycle in tumorigenesis of endometrial cancer [26]. Another recent paper had as its topic the effect of MSX1, a transcription repressor, on the cell cycle. In this paper, Yue et al. postulated that MSX1 induces G0/G1 cell-cycle-arrest in cervical cancer cells [27], so for us, MSX1 was an interesting marker to analyze.

The analysis of the tumor material significantly showed that endometrioid endometrial carcinomas had the strongest staining intensity of MSX1. The tumors with the worst grading (grading 3, poorly differentiated) showed mostly no MSX1-expression at all, while graded-2 tumors showed the strongest intensity. Tumors staged as T1 showed significantly more staining than tumors with higher T statuses. When we investigated the lymph node status, we found that most patients who showed a strong intensity of MSX1-staining did not have any involvement of the lymph nodes. On the other hand, patients with infiltrated lymphoid nodes showed a much weaker staining intensity or no staining at all.

In a study by Huang et al. the research team established a five-gene biomarker panel for predicting lymph node metastasis in patients with early stage endometrial cancer [31]. The research team investigated on eight genes and one clinical feature (depth of myometrial invasion), with MSX1 being one of the genes. In the end, five of the genes made it on the panel. Since the results were not significant enough, the MSX1-gene was not one of them. Our results on the other hand, significantly indicate a connection between the MSX1-expression and the lymph node status. Whether there is a causal connection may need further investigation. The main difference between this paper and our research is that we looked at the MSX1 protein expression, whereas Huang et al. analyzed the genome.

In our study, we found significant correlations between MSX1 and β-Catenin, p53, p21, and the steroid receptors ERα, ERβ, PRα, and PRβ. These correlations are in accordance with the results of previous studies by other research groups [32,33,34,35,36]. Our results confirm these findings in the sense that our correlations are significant (*p* < 0.05) and have a high correlation coefficient (cc). Figure 6 is a simplified model of how these proteins interact as well as MSX1’s influence on the cell cycle. MSX1 is a target gene of β-Catenin and the estrogen receptors and it has a stabilizing effect on p53, which activates p21 (p53/p21 pathway).

Yue et al. found out that MSX1 has tumor-suppressive functions by inducing apoptosis and cell cycle arrest in breast cancer tumorigenesis, so they postulate that a methylation of MSX1 can be used a biomarker for early diagnosis and detection of breast cancer [25]. We postulate that MSX1 could be an interesting biomarker not only for breast cancer but also for other gynecological tumors like in our case for endometroid cancer.

We were also able to show, that expression of the protein MSX1 correlates with the length of survival in patients with endometrioid and clear-cell carcinomas of the endometrium and the ovary: The subgroup of patients with endometrioid endometrial carcinomas and an MSX1 expression of more than 10% showed a median survival of 21.3 years. In comparison, patients with less or no expression at all had a median survival of 17.3 years. A longer survival implies a higher oncologic security and may support—within strict limitations—the possibility of a transient uterus-preserving therapy.

The guidelines for fertility-preserving therapy in endometrial carcinomas are recommended for endometrioid adenocarcinomas cT1A, G1 without infiltration of the myometrium, and with the expression of progesterone receptors. Uterus and adnexa can be preserved if a hysteroscopy with biopsy or abrasion to confirm the diagnosis has been executed. An infiltration of the adnexa or the myometrium must be ruled out by laparoscopy with vaginal ultrasound or MRT. The patient must receive systemic therapy with MPA, MGA, or LNG-IUD and the response must be controlled after 6 months by hysteroscopy with abrasion. If there is a complete remission, a pregnancy can be pursued. After childbirth or resignation, a total hysterectomy with extirpation of both adnexa is strongly recommended [1].

## 4. Materials and Methods

### 4.1. Patients and Tissue Collection

For this study, samples of endometrioid and clear cell endometrial carcinomas and endometrioid and clear cell ovarian carcinomas were used. A total of 97 patients was analyzed retrospectively.

The samples were taken from the hospital archive of the Department of Pathology, Ludwig-Maximilians-University. Surgery took place between 1 January 1990 and 31 December 2001 in the Department of Gynecology, Ludwig-Maximilians-University Munich, Germany. The patients were staged and the tumors were graded according to the criteria of the 1988 International Federation of Gynecology and Obstetrics (FIGO) [38]. They were all formalin-fixed and paraffin-embedded. Endometrioid endometrial carcinoma tissues (*n* = 59), clear cell endometrial carcinoma tissues (*n* = 6), endometrioid ovarian carcinoma tissues (*n* = 21) and clear cell ovarian carcinoma tissues (*n* = 11) were used. The tumor register of Munich provided the data about the patients’ survival. Table 3 and Table 4 show patients’ characteristics. Information regarding survival includes the date of confirmed histological diagnosis after primary surgery to the date of recurrence or last visit.

### 4.2. Immunohistochemistry

For the immunohistochemical staining of the endometrial and ovarian carcinomas, tissue microarrays (tma) were used. Therefore, new samples of the original slides were taken and representative areas, where tumor was expected, were selected. After that a pathologist verified the presence of tumor tissue on the samples.

The paraffinized tma-slides were put into Roticlear, a substitute-medium for xylol, for 20 min to deparaffinize the tissue. After that, they were put into 100% ethanol and then for another 20 min into 3% hydrogen peroxide diluted in methanol to inhibit the endogenous peroxidase activity. A series of graded alcohols (100%, 70%, 50%) and eventually distilled water was used to rehydrate the samples. The slides were demasked by heating them up to 100 °C in a pressure cooker, containing an already boiling trisodium citrate buffer solution with pH = 6.0 and cooked for 5 min.

After that a blocking solution (Reagent 1 of polymer detection kit (ZytoChem Plus HRP Polymer System, Mouse Rabbit, Zytomed, Berlin, Germany)) was used to saturate the electrostatic charges in the tissue and the primary antibody was placed on the samples and incubated for 16 h at 4 °C. The antibody used was Anti-MSX1, Rabbit IgG polyclonal, concentration 0.2 mg/mL (Sigma; order number HPA073604. Sigma-Aldrich, Merck KGaA, Darmstadt, Germany), which was diluted at a ratio of 1:200 with PBS. The next day the antibody-surplus was washed off and a post-block-reagent applied for 20 min. Secondary antibodies were conjugated with horseradish peroxidase (HRP) for another 30 min, and finally the antibody was stained, using DAB (chromogen substrate kit, Dako North America Inc., Carpinteria, CA, USA) for 5 min. The staining reaction was then stopped with distilled water and the material was counterstained using hematoxylin. Finally, the samples were dehydrated using an ascending series of graded alcohols (50%, 70%, 96%, and 100%) and Roticlear. As a last step, the samples were covered with a mounting medium and cover glasses.

For evaluation, two independent blinded observers looked at the staining intensity (0 = no staining, 1 = weak intensity, 2 = moderate intensity, 3 = strong intensity), and the percentage of stained carcinoma cells. The evaluation of the two observers differed in 4 cases (*n* = 4.1%) Both observers reevaluated those cases together and came to the same results. The concordance before the reevaluation was 95.9%. Also the IRS (immunoreactive score) was applied, which is the result of the percentage of stained carcinoma cells (0 = 0%, 1 = 1–10%, 2 = 11–50%, 3 = 51–80%, 4 ≤ 81%) multiplied with the coded staining intensity [28]. In statistical analysis, all three results were looked at separately (staining intensity, percentage of stained carcinoma cells, and IRS).

As we are trying to establish a procedure, a routine pathology can perform, we concentrated on immunohistochemistry and did not perform promoter’s methylation analysis. As several studies suggest, promoter methylation of the *MSX1* gene is presumable for this kind of tumor [21,24,25].

### 4.3. Statistical Analysis

The statistical analysis software SPSS (IBM SPSS Statistics, Version 25; IBM Deutschland GmbH, Ehningen, Germany) was used. For evaluation of the clinical-pathological variables, the Kruskal–Wallis test was applied. Survival data were rendered using the Kaplan–Meier analysis and correlations were tested, using the bivariate correlation of Spearman.

### 4.4. Ethics Approval

This study was conducted conforming to the Declaration of Helsinki 1975 and it was approved by the Ethics Committee of the Ludwig-Maximilians-University, Munich, Germany (approval number 449-14, 14.03.2014). All patients’ data were fully anonymized, and during experimental analysis, the authors were blinded for clinical information. All tumor tissue used was leftover material and all diagnostic procedures had already been completed, when the samples were received for the study.

## 5. Conclusions

To summarize, it can be said that an MSX1-expression can be found mainly in endometrioid endometrial carcinomas and is stronger stained in tumor material of patients with positive prognostic variables like a favorable T status or N status. Also, in grading 2 tumors, a significantly higher IRS was observed. Positive correlations were found to the markers β-Catenin, p53, p21, and the steroid receptors ERα, ERβ, PRα, and PRβ. Patients with an MSX1-expression in more than 10% of tumor cells had a significant longer survival compared to those without this expression-pattern. This makes MSX1 a significant marker for better survival and may allow, under strict conditions, a uterus-sparing therapy in patients with endometrial cancer with a higher oncologic security.

## Figures and Tables

**Figure 1 ijms-21-04529-f001:**
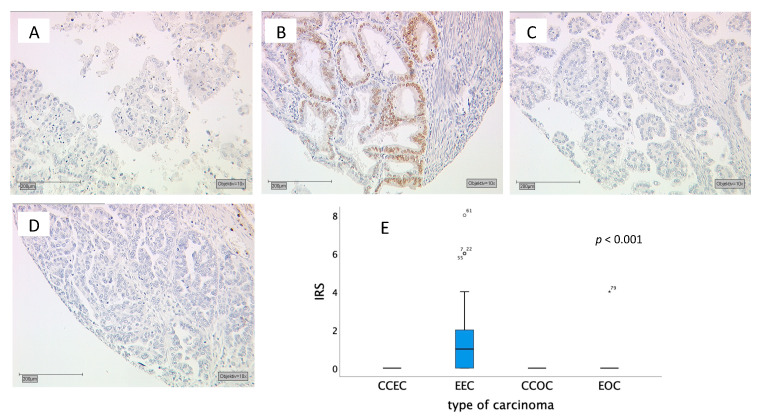
(**A**–**D**) Tumor samples stained for MSX1: (**A**) Clear cell endometrial carcinoma with an immunoreactive score (IRS) of 0; (**B**) Endometrioid endometrial Carcinoma with an IRS of 2; (**C**) Clear cell ovarian carcinoma with an IRS of 0; (**D**) Endometrioid ovarian carcinoma with an IRS of 0; (**E**) Boxplots with a median IRS of 1 for endometrioid endometrial carcinomas (EEC) and with a median IRS of 0 for clear cell endometrial carcinomas (CCEC), clear cell ovarian carcinomas (CCOC) and endometrioid ovarian carcinomas (EOC). The circle indicates values more than 1.5 box lengths from the 75th percentile. Asterisks indicate values which are more than 2times the box length. Numbers at circle and asterisks indicate sample number.

**Figure 2 ijms-21-04529-f002:**
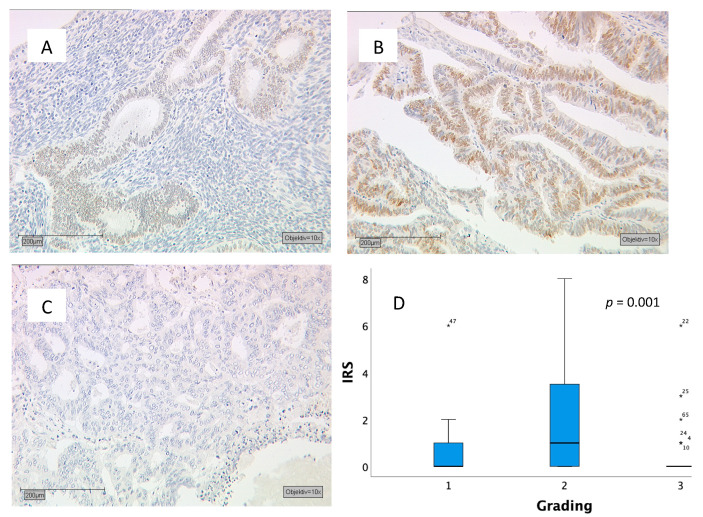
(**A**–**C**) Tumor samples stained for MSX1: (**A**) Grading 1 tumor with an IRS of 2 (endometrioid ovarian carcinoma); (**B**) Grading 2 tumor with an IRS of 6 (endometrioid endometrial carcinoma); (**C**) Grading 3 tumor with an IRS of 0 (endometrioid endometrial carcinoma); (**D**) Boxplots with a median IRS of 1 for grading 2 tumors (moderately differentiated) and with a median IRS of 0 for grading 1 (well-differentiated) and grading 3 (badly differentiated) tumors. Asterisks indicate values which are more than 2times the box length. Numbers at asterisks indicate sample number.

**Figure 3 ijms-21-04529-f003:**
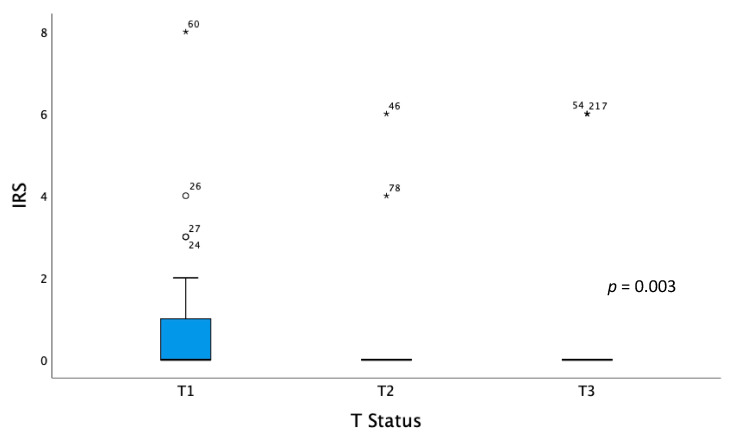
Boxplots for MSX1-staining in tumors with T1, T2, and T3 staging. The circle indicates values more than 1.5 box lengths from the 75th percentile. Asterisks indicate values which are more than 2times the box length. Numbers at circle and asterisks indicate sample number.

**Figure 4 ijms-21-04529-f004:**
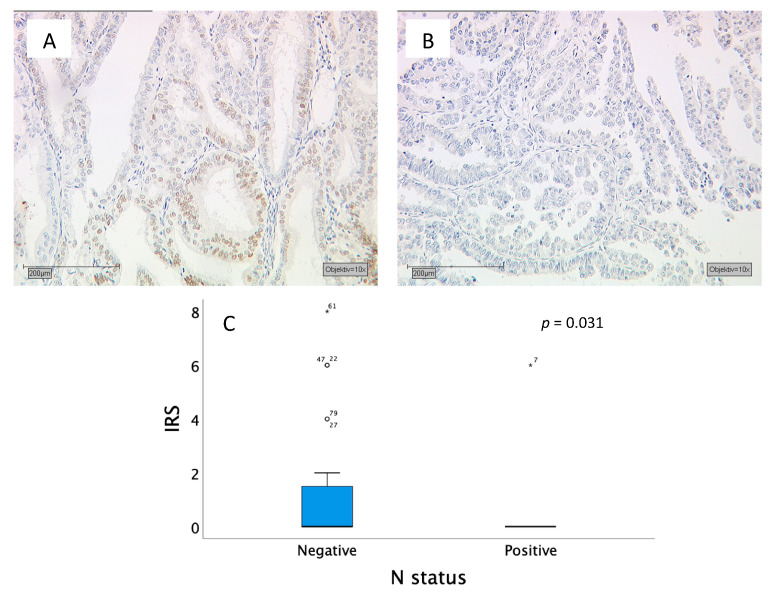
(**A**) and (**B**) Tumor samples stained for MSX1: (**A**) Tumor material of a patient with negative N status with an IRS of 2; (**B**) Tumor material of a patient with positive N status with an IRS of 0; (**C**) Boxplots with a median IRS of 0 for negative N status and with a median IRS of 0 for positive N status. The circle indicates values more than 1.5 box lengths from the 75th percentile. Asterisks indicate values which are more than 2times the box length. Numbers at circle and asterisks indicate sample number.

**Figure 5 ijms-21-04529-f005:**
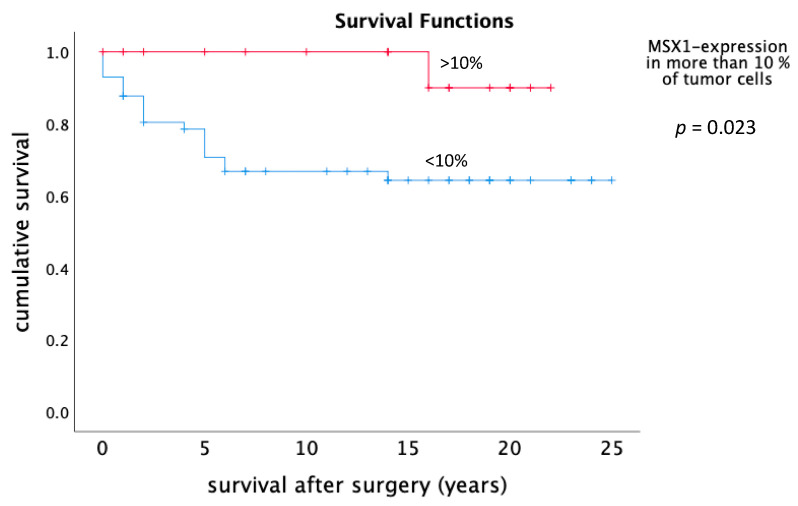
Survival of patients with MSX1-expression in more than 10% of the tumor cells and less than 10% MSX-1 expression or no expression.

**Figure 6 ijms-21-04529-f006:**
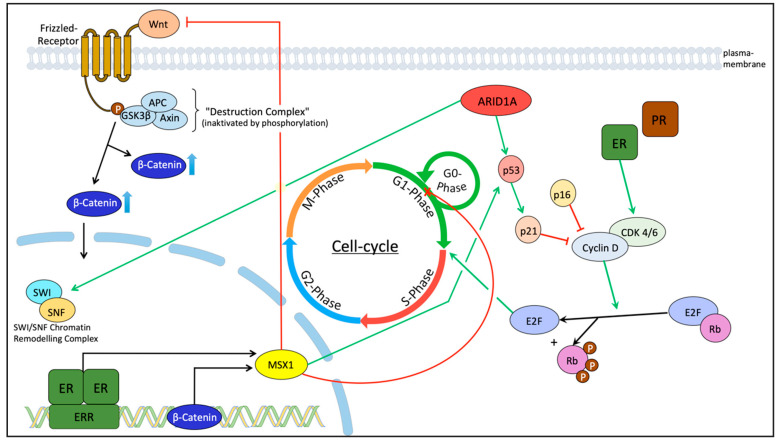
The function of MSX1 in relation to prognostic markers found in our former investigation and investigations from other groups [32,33,34] (basic structure of the figure based on Katarzyna et al. [26]): MSX1 is a known inhibitor of the Wnt-signaling [25] and the result of the Wnt-signaling is the shuttling of β-Catenin from the cytoplasm to the nucleus [37]. Based on this figure, we found a positive correlation of MSX1 and β-Catenin in the cytoplasm. (Destruction-Complex (DC): inactivated by Wnt via Frizzled-receptor (can be inactivated in tumor cells). β-Catenin: accumulates, when DC is inactivated (leads to transcription of target genes (MSX1). SWI/SNF-Chromatin-Remodeling-Complex: acts as tumor suppressor; needs ARID1A. ER: Estrogen receptor; PR: Progesterone receptor; E2F: E2-transcriptional factor; Rb: Retinoblastoma). Green arrows describe a stimulating effect, red arrows describe an inhibitory effect.

**Table 1 ijms-21-04529-t001:** Multivariate Cox regression analysis for MSX1.

Variables	Hazard Ratio	95% Confidence Interval	*p*-Value
MSX1 > 10%	0.147	0.019–1.139	0.066
Age at diagnosis	1.023	0.974–1.075	0.335
Grading	0.851	0.475–1.524	0.587
Clear-cell vs. Endometrioid	0.555	0.173–1.780	0.322
FIGO	0.969	0.664–1.416	0.871
p53RS	0.509	0.182–1.428	0.200
β-Catenin RS	1.027	0.419–2.739	0.885

**Table 2 ijms-21-04529-t002:** Positive correlations between MSX1 and formerly investigated markers.

Protein	β-Catenin	p21	p53	ERα	ERβ	PRα	PRβ
Variable	βCatenin RS	cell-count	p53RS	IRS	IRS	IRS	IRS
MSX1	intensity	percentage	percentage	percentage	intensity	IRS	intensity
cc	0.427	0.408	0.343	0.418	0.439	0.395	0.49
*p*	<0.001	<0.001	0.002	0.003	0.001	0.004	<0.001
*n*	80	81	80	50	52	52	52

cc: correlation coefficient; βCatenin RS: 0 = membrane negative, 1 = membrane positive, 2 = nucleus positive; p53RS: 0 = negative/strongly positive, 1 = positive; cell-count: 0 = 0, 1 = < 10%, 2 = > 10% < 50%.

**Table 3 ijms-21-04529-t003:** Patients’ characteristics (endometrial carcinoma, *n* = 65).

Age (Median)	63.7 (Range 35–82)
Clear cell uterine carcinoma	6 (9.2)
Endometrioid uterine carcinoma	59 (90.8)
FIGO-Staging	
I	12 (18.5)
II	13 (20)
III	18 (27.7)
IV	22 (33.8)
Grading	
Grade 1	18 (27.7)
Grade 2	24 (36.9)
Grade 3	23 (35.4)
Tumor size	
pT1	50 (76.9)
pT2	7 (10.8)
pT3	8 (12.3)
pT4	0 (0)
Patients with	
Diabetes	8 (12.3)
Hypertonus	16 (24.6)
Adipositas	23 (35.3)
None	18 (27.7)
Progression (over 177 months)	
None	51 (78.5)
At least one	11 (16.9)
Not available	3 (4.6)
Survival (over 177 months)	
Right censured	32 (49.2)
Died	32 (49.2)
Not available	1 (1.5)
Patients who received surgery	65 (100)
Patients who received radiation/chemotherapy	9 (13.8)
Patients who denied radiation therapy	1 (1.5)

**Table 4 ijms-21-04529-t004:** Patients’ characteristics (ovarian carcinoma, *n* = 32).

Age (Median)	56.9 (Range 42–77)
Clear cell ovarian carcinoma	11 (34.4)
Endometrioid ovarian carcinoma	21 (65.6)
FIGO-Staging	
I	12 (37.5)
II	7 (21.9)
III	6 (18.8)
IV	7 (21.9)
Grading	
Grade 1	6 (18.8)
Grade 2	5 (15.6)
Grade 3	19 (59.4)
Tumor size	
pT1	14 (43.8)
pT2	6 (18.8)
pT3	10 (31.3)
pT4	2 (6.3)

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
