# Peer review of "MSX1—A Potential Marker for Uterus-Preserving Therapy of Endometrial Carcinomas"

_ijms, 2020, doi:10.3390/ijms21124529_

Round 1

Reviewer 1 Report

This manuscript primarily describes MSX1 expression correlates with endometrioid endometrial carcinomas, grading 2, and improved long-term survival. While the paper is of interest to the field and generally well written there are several major issues which need to be addressed:

Major issues

  1. The images in the IHC article do not show intense nuclear staining as seen with the same antibody in the Human Protein Atlas database (https://www.proteinatlas.org/ENSG00000163132 MSX1/pathology/endometrial+cancer). The authors observe some differences in expression levels with their series of endometrial and ovarian cancers, but the observed IRS values are very low. These low IRS values can produce variations among observers. It is necessary to carry out a study Interobserver variability for MSX 1 score was calculated using Cohen κ.
  2. In previous works (Bonito et al. 2016, Yue et al. 2018) the MSX1 promoter's methylation is always evaluated as a gold standard. The authors should validate their IHC analysis with a correlation to the promoter's methylation analysis so that they can demonstrate that their antibody is effective in assessing loss of MSX1 expression by IHC.
  3. The statistical analyses performed are correct, but they should perform univariate and multivariate analyses of the prognostic factors in the different grades of endometrium analyzed. On the other hand, analysis of MSX1 at sensitivity and specificity levels has not been performed, as well as analysis of ROC curve.
  4. EEC grade 2 and grade 1 present the most variable expression, as shown in figure 2, is where it would be more interesting to see the ability to associate the MSX1 expression with overall survival. Kaplan Meier analysis should be performed with the data from these grades and indicating whether they are early or advanced stages.

Minor issues

  1. Images with intense nuclear staining should be added to the figures.
  2. Adding a table with patient data would improve the ability to follow the series in Kaplan Meier analysis.

Author Response

Reviewer 1:

This manuscript primarily describes MSX1 expression correlates with endometrioid endometrial carcinomas, grading 2, and improved long-term survival. While the paper is of interest to the field and generally well written there are several major issues which need to be addressed:

Major issues

    1. The images in the IHC article do not show intense nuclear staining as seen with the same antibody in the Human Protein Atlas database (https://www.proteinatlas.org/ENSG00000163132MSX1/pathology/endometrial+cancer). The authors observe some differences in expression levels with their series of endometrial and ovarian cancers, but the observed IRS values are very low. These low IRS values can produce variations among observers. It is necessary to carry out a study Interobserver variability for MSX 1 score was calculated using Cohen κ.

      Thank you for this remark, we added information on the interobserver variability and changed the corresponding paragraph accordingly:

      "For evaluation two independent blinded observers looked at the staining intensity (0 = no staining, 1 = weak intensity, 2 = moderate intensity, 3 = strong intensity), and the percentage of stained carcinoma cells. The evaluation of the two observers differed in 4 cases (n = 4.1%) Both observers reevaluated those cases together and came to the same results. The concordance before the reevaluation was 95.9%."

  • In previous works (Bonito et al. 2016, Yue et al. 2018) the MSX1 promoter's methylation is always evaluated as a gold standard. The authors should validate their IHC analysis with a correlation to the promoter's methylation analysis so that they can demonstrate that their antibody is effective in assessing loss of MSX1 expression by IHC.

    This is a good point. Unfortunately we were not able to perform promoter's methylation analysis. Reason for this is the time-consuming procedure which would not allow us to adhere to the deadline set by the journal.
    Idea of this paper was that a routine pathology could detect a group of patients, that would benefit from less invasive procedures, using only immunohistochemistry. For this reason we did not include promoter's methylation analysis in the first place. Foundation for this approach are previous works done by other research groups, which describe the connection between promoter’s methylation of the MSX1-gene and several types of cancer
    [1-3]. We added two sentences to the section "4.2. Immunohistochemistry" to clarify this.

    "
    As we are trying to establish a procedure, a routine pathology can perform, we concentrated on immunohistochemistry and did not perform promoter’s methylation analysis. As several studies suggest, promoter methylation of the MSX1 gene is presumable for this kind of tumor [21,24,25]."

  • The statistical analyses performed are correct, but they should perform univariate and multivariate analyses of the prognostic factors in the different grades of endometrium analyzed. On the other hand, analysis of MSX1 at sensitivity and specificity levels has not been performed, as well as analysis of ROC curve.

    Thank you for this idea. We performed Kaplan Meier analysis for the different grades of endometrial cancer. Unfortunately no significant results were observed.Please find the corresponding Kaplan Meier overall survival curves in the supplementary PDF file Fig. 7,8,9.

    We also performed multivariate analysis using cox-regression and added the results to our paper under "2.3 Multivariate analysis":

    "For multivariate analysis we used Cox regression with tumor-related death as end point. Our findings do not show any significant results. The lowest p-value can be found for the variable "MSX1 >10%" (MSX1-expression in more than 10% of tumor cells) (p=0.066) with a hazard ratio of 0.147. The exact results are listed in Table 1."

    We also included a new table (Table 1) displaying the detailed results. Unfortunately no significant results were observed. even though MSX1 >10%" (MSX1-expression in more than 10% of tumor cells) had the lowest p-value (p=0.066) with a hazard ratio of 0.147.

    Table 1:

Variables

Hazard Ratio 

95% Confidence Interval

p-Value

MSX1 >10%

0.147

0.019-1.139

0.066

Age at diagnosis

1.023

0.974-1.075

0.335

Grading

0.851

0.475-1.524

0.587

clear-cell vs endometrioid

0.555

0.173-1.780

0.322

FIGO

 0.969

0.664-1.416

0.871

p53RS

0.509

0.182-1.428

0.200

β-CateninRS

1.027

0.419-2.739

0.885

In ROC curve analysis, we evaluated the IRS and the percentage of MSX1 expression in tumor cells. Neither analysis showed significance: for the IRS the p-value lies at 0.619 with an intervall of confidence of 0.390 to 0.686, for the percentage the p-value lies at 0.591 with an intervall of confidence of 0.394 to 0.688. The corresponding ROC curves can be found in the supplementary PDF-file Fig. 10,11.

  1. EEC grade 2 and grade 1 present the most variable expression, as shown in figure 2, is where it would be more interesting to see the ability to associate the MSX1 expression with overall survival. Kaplan Meier analysis should be performed with the data from these grades and indicating whether they are early or advanced stages.

    We performed Kaplan Meier analysis with the data from all three grades, but unfortunately could not find any significant results. Please find the corresponding Kaplan Meier overall survival curves in the supplementary PDF-file Fig. 7,8,9.

Minor issues

  1. Images with intense nuclear staining should be added to the figures.

    Images with higher intensity in nuclear staining were added to the figures.

  1. Adding a table with patient data would improve the ability to follow the series in Kaplan Meier analysis.

    Information on clinical parameters was displayed in table 2. We added more information to this table and separated it into two: Table 3 now displays extended information regarding the group of patients with endometrial carcinomas, while table 4 now shows information on the group of patients with ovarian carcinomas.
    Table 3:

Age (median)

63.7 (range 35-82)

Tumor-subtype:

Clear cell uterine carcinoma

6 (9.2)

Endometrioid uterine carcinoma

59 (90.8)

FIGO-Staging:

I

12 (18.5)

II

13 (20)

III

18 (27.7)

IV

22 (33.8)

Grading:

Grade 1

18 (27.7)

Grade 2

24 (36.9)

Grade 3

23 (35.4)

Tumor size

pT1

50 (76.9)

pT2

7 (10.8)

pT3

8 (12.3)

pT4

0 (0)

Patients with

Diabetes

8 (12.3)

Hypertonus

16 (24.6)

Adipositas

23 (35.3)

None

18 (27.7)

Progression (over 177 months)

None

51 (78.5)

At least one

11 (16.9)

Not available

3 (4.6)

Survival (over 177 months)

Right censured

32 (49.2)

Died

32 (49.2)

Not available

1 (1.5)

Patients who received surgery

65 (100)

Patients who received radiation/chemotherapy

9 (13.8)

Patients who denied radiation therapy

1 (1.5)

Table 4:

Age (median)

56.9 (range 42-77)

Tumor-subtype:

Clear cell ovarian carcinoma

11 (34.4)

Endometrioid ovarian carcinoma

21 (65.6)

FIGO-Staging:

I

12 (37.5)

II

7 (21.9)

III

6 (18.8)

IV

7 (21.9)

Grading:

Grade 1

6 (18.8)

Grade 2

5 (15.6)

Grade 3

19 (59.4)

Tumor size

pT1

14 (43.8)

pT2

6 (18.8)

pT3

10 (31,3)

pT4

2 (6.3)

Reviewer 2 Report

The manuscript entitled “MSX1 - a potential marker for uterus-preserving therapy of endometrial carcinomas” by Simon Eppich et al is focused on a new potential prognostic marker, a relevant topic in the field of in gynaecological tumors.

The manuscript investigates  the  expression of the protein MSX1, a transcription repressor, which has an inhibitory effect on the cell cycle in a retrospective cohort of endometrioid endometrial carcinomas, clear cell endometrial carcinomas, endometrioid ovarian carcinomas and clear cell ovarian carcinomas. The authors demonstrated that MSX1 expression correlates with improved long-term survival in gynaecological cancers.

Major revisions

  • Please stratify patients in Kaplan Meier overall survival curves in terms of IRS (immunoreactive score).
  • Please add independent Kaplan Meier overall survival curves for endometrial carcinomas (endometrioid endometrial carcinomas+ clear cell endometrial carcinomas) and for ovarian carcinoma (endometrioid ovarian carcinomas and clear cell ovarian carcinomas).
  • Please indicate the type of treatment (both in terms of surgery and chemoradiotherapy) of patients enrolled in the study.

Minor revisions

  • Please clarify the choice of cut-off of 10% MSX1 expression staining to stratify patients in Kaplan Meier overall survival curve.
  • Please introduce in Table 2 more clinical parameters regarding the patients enrolled in the study.
  • It could would be useful to introduce in the manuscript data relative to ki67 staining relative to patients enrolled in the study.

Author Response

Reviewer 2:

The manuscript entitled “MSX1 - a potential marker for uterus-preserving therapy of endometrial carcinomas” by Simon Eppich et al is focused on a new potential prognostic marker, a relevant topic in the field of in gynaecological tumors.

The manuscript investigates the expression of the protein MSX1, a transcription repressor, which has an inhibitory effect on the cell cycle in a retrospective cohort of endometrioid endometrial carcinomas, clear cell endometrial carcinomas, endometrioid ovarian carcinomas and clear cell ovarian carcinomas. The authors demonstrated that MSX1 expression correlates with improved long-term survival in gynaecological cancers.

Major revisions

       
Please stratify patients in Kaplan Meier overall survival curves in terms of IRS (immunoreactive score).

Unfortunately no significant results were found for Kaplan Meier overall survival in terms of IRS. Please find the corresponding Kaplan Meier curves in the supplementary PDF-file Fig. 1,2,3,4.

       
Please add independent Kaplan Meier overall survival curves for endometrial carcinomas (endometrioid endometrial carcinomas+ clear cell endometrial carcinomas) and for ovarian carcinoma (endometrioid ovarian carcinomas and clear cell ovarian carcinomas).

We performed Kaplan Meier overall survival curves for endometrial carcinomas and for ovarian carcinomas. Unfortunately for those entities no significant results can be reported. Please find the Kaplan Meier overall survival curves in the supplementary PDF-file Fig. 5,6.

       
Please indicate the type of treatment (both in terms of surgery and chemoradiotherapy) of patients enrolled in the study.

We added information on patients' surgery, chemotherapy, and radiotherapy to table 2 (view below).

Minor revisions

       
Please clarify the choice of cut-off of 10% MSX1 expression staining to stratify patients in Kaplan Meier overall survival curve.

As a ROC curve analysis did not identify a suitable cut-off (supplementary PDF-file) the idea was to look at the percentage distribution of type of tumor. There the endometrioid carcinomas had their boxplot displaying the 0-50th percentile is located at the 10% mark, which gave the impetus for this cut-off. We added the following sentences:

"
Reason for this cut-off was the percentage distribution of type of tumor. When looking at endometrioid carcinomas the boxplot displaying the 0-50th percentile is located at the 10% mark."

       
Please introduce in Table 2 more clinical parameters regarding the patients enrolled in the study.

      To include more data and to preserve clarity, we made two separate tables out of table 2 (table 3 and table 4). Table 3 now displays the clinical parameters of the patients with endometrial carcinomas, while table 4 displays the clinical parameters of the patients with ovarian carcinomas. We hope, that this additional data will be sufficient.

Table 3:

Age (median)

63.7 (range 35-82)

Tumor-subtype:

Clear cell uterine carcinoma

6 (9.2)

Endometrioid uterine carcinoma

59 (90.8)

FIGO-Staging:

I

12 (18.5)

II

13 (20)

III

18 (27.7)

IV

22 (33.8)

Grading:

Grade 1

18 (27.7)

Grade 2

24 (36.9)

Grade 3

23 (35.4)

Tumor size

pT1

50 (76.9)

pT2

7 (10.8)

pT3

8 (12.3)

pT4

0 (0)

Patients with

Diabetes

8 (12.3)

Hypertonus

16 (24.6)

Adipositas

23 (35.3)

None

18 (27.7)

Progression (over 177 months)

None

51 (78.5)

At least one

11 (16.9)

Not available

3 (4.6)

Survival (over 177 months)

Right censured

32 (49.2)

Died

32 (49.2)

Not available

1 (1.5)

Patients who received surgery

65 (100)

Patients who received radiation/chemo therapy

9 (13.8)

Patients who denied radiation therapy

1 (1.5)

Table 4:

Age (median)

56.9 (range 42-77)

Tumor-subtype:

Clear cell ovarian carcinoma

11 (34.4)

Endometrioid ovarian carcinoma

21 (65.6)

FIGO-Staging:

I

12 (37.5)

II

7 (21.9)

III

6 (18.8)

IV

7 (21.9)

Grading:

Grade 1

6 (18.8)

Grade 2

5 (15.6)

Grade 3

19 (59.4)

Tumor size

pT1

14 (43.8)

pT2

6 (18.8)

pT3

10 (31,3)

pT4

2 (6.3)

       It could would be useful to introduce in the manuscript data relative to ki67 staining relative to patients enrolled in the study.

Thank you for this interesting thought. Unfortunately, we do not have information on ki67 with this cohort of patients. This could be the subject of future research.

References:

  1. Shames, D.S.; Girard, L.; Gao, B.; Sato, M.; Lewis, C.M.; Shivapurkar, N.; Jiang, A.; Perou, C.M.; Kim, Y.H.; Pollack, J.R., et al. A genome-wide screen for promoter methylation in lung cancer identifies novel methylation markers for multiple malignancies. PLoS medicine 2006, 3, e486, doi:10.1371/journal.pmed.0030486.
  2. Dunwell, T.L.; Hesson, L.B.; Pavlova, T.; Zabarovska, V.; Kashuba, V.; Catchpoole, D.; Chiaramonte, R.; Brini, A.T.; Griffiths, M.; Maher, E.R., et al. Epigenetic analysis of childhood acute lymphoblastic leukemia. Epigenetics 2009, 4, 185-193.
  3. Yue, Y.; Yuan, Y.; Li, L.; Fan, J.; Li, C.; Peng, W.; Ren, G. Homeobox protein MSX1 inhibits the growth and metastasis of breast cancer cells and is frequently silenced by promoter methylation. Int J Mol Med 2018, 41, 2986-2996, doi:10.3892/ijmm.2018.3468.

Round 2

Reviewer 1 Report

The authors have correctly answered all the suggestions made in the previous version.